# Edge Caching Data Distribution Strategy with Minimum Energy Consumption

**DOI:** 10.3390/s24092898

**Published:** 2024-05-01

**Authors:** Zhi Lin, Jiarong Liang

**Affiliations:** School of Computer, Electronics and Information, Guangxi University, Nanning 530004, China; 13977106752@163.com

**Keywords:** mobile edge computing, edge caching, green communication, mobile communication, greedy algorithm

## Abstract

In the context of the rapid development of the Internet of Vehicles, virtual reality, automatic driving and the industrial Internet, the terminal devices in the network show explosive growth. As a result, more and more information is generated from the edge of the network, which makes the data throughput increase dramatically in the mobile communication network. As the key technology of the fifth-generation mobile communication network, mobile edge caching technology which caches popular data to the edge server deployed at the edge of the network avoids the data transmission delay of the backhaul link and the occurrence of network congestion. With the growing scale of the network, distributing hot data from cloud servers to edge servers will generate huge energy consumption. To realize the green and sustainable development of the communication industry and reduce the energy consumption of distribution of data that needs to be cached in edge servers, we make the first attempt to propose and solve the problem of edge caching data distribution with minimum energy consumption (ECDDMEC) in this paper. First, we model and formulate the problem as a constrained optimization problem and then prove its NP-hardness. Subsequently, we design a greedy algorithm with computational complexity of $O(n^{2})$ to solve the problem approximately. Experimental results show that compared with the distribution strategy of each edge server directly requesting data from the cloud server, the strategy obtained by the algorithm can significantly reduce the energy consumption of data distribution.

## 1. Introduction

The rapid advancement of mobile communication and Internet of Things (IoT) technologies has resulted in an exponential increase in the scale of mobile communication networks and the proliferation of mobile terminal devices which will generate a massive volume of data, leading to significant network traffic and congestion. Projections indicate that by 2025 the aggregate number of diverse terminal devices connected to these networks will soar to 34.1 billion and the global data volume will increase to 175 ZB [1,2]. The emergence of latency-sensitive applications, such as virtual reality, augmented reality, autopilot systems, and real-time navigation, further exacerbates this issue. These applications demand high levels of real-time responsiveness, ultra-low latency, and substantial network throughput. The traditional centralized network architecture, predominantly based on cloud computing paradigms, is ill-equipped to meet the stringent requirements of end-users in terms of these performance metrics [3,4].

Mobile edge computing (MEC) addresses the challenge of ensuring low latency, a limitation often encountered in cloud computing [5]. In the MEC framework, edge servers are deployed at base stations or wireless network access points located in close proximity to end-user terminals [5,6]. This configuration enables application providers to lease computing, caching, and communication resources on edge servers, facilitating the hosting of data-intensive applications and delivering services with low latency to end-users [3,7]. By caching application data on the edge server, not only can the latency in data retrieval and transmission be significantly reduced, but it also contributes to a decrease in traffic on the backbone network and content distribution network, thereby mitigating network congestion [1,8,9]. It is essential to note, however, that the storage capacity of edge servers is constrained, leading to the caching of only highly popular data [10,11,12]. This selective caching approach ensures optimal resource utilization.

In recent years, numerous scholars have conducted extensive research on edge caching from various perspectives. A predominant focus in the field of edge caching has been the reduction in data transmission delays. Ref. [13] explored the edge caching problem in both single-slot and multi-slot scenarios. Leveraging multi-agent reinforcement learning technology, the study in [13] minimized user data acquisition delays. In [14], the investigation incorporated considerations for device-to-device (D2D) communication interference. To mitigate delays, an optimization algorithm for a probabilistic caching strategy was proposed, resulting in a substantial reduction in data transmission delays.

Reducing the cost of edge caching and enhancing the revenue generated by edge caching represents another significant research orientation in this domain. In [15], Abolhassani et al. introduced a caching strategy driven by data freshness to tackle the caching challenges posed by dynamically changing content in distributed edge caching and single edge caching scenarios. The devised strategy effectively optimized cache space and reduced the average cache cost of the system. Ref. [16] jointly considered the freshness, popularity, and similarity of content, proposing a cost-effective edge caching algorithm termed SAPoC. Additionally, from the perspective of application providers. In [17] Xia et al. presented an online mobile edge data caching method known as OL-MEDC, aimed at maximizing cache revenue in mobile edge computing systems. Experimental results demonstrated the method’s efficiency, reliability, and superior performance compared to existing conventional approaches, showcasing its suitability for large-scale deployment in practical edge computing environments. Moreover, energy consumption cost is an integral component of edge caching expenses. To address the cache energy efficiency problem in edge computing, Xu et al. [18] modeled the edge computing scenario as a three-layer heterogeneous network structure comprising the server layer, edge layer, and user layer. Subsequently, they considered distinct data lifetimes and designed two energy-efficient cache methods with in-memory storage and processing technology.

The consideration of data popularity holds significant weight in the investigation of edge caching; taking into account both data popularity and freshness in the named data network, Alduayji et al. [19] proposed a scheme named PF EdgeCache. The experimental results of [19] illustrated that the proposed scheme outperformed previously proposed cache schemes in terms of simplicity and effectiveness. In [20], considering a two-layer heterogeneous network comprising macro base stations and micro base stations, the authors proposed a layered edge cache scheme based on deep learning. Initially, they introduced a novel content popularity prediction method using the stacked autoencoder long short term memory network (SAE-LSTMNet), characterized by high accuracy and short execution time. Subsequently, leveraging the predicted content popularity, they formulated a layered edge caching problem which is NP-hard. Finally, an approximate algorithm with low computational complexity is devised to address this problem.

The security of data in edge caching systems has emerged as a prominent area of research. In [21], the authors introduced blockchain technology and then they proposed a decentralized and secure data caching strategy and transaction scheme. This proposed scheme not only reduced data transmission delays but also enhanced data cache hit rates and the transaction rate of data transactions while safeguarding against data leaks. Additionally, in [22] the authors proposed a secure edge caching scheme based on reinforcement learning principles. The scheme offered lightweight authentication mechanisms and safeguarded user’s privacy against potential eavesdropping threats. Furthermore, Wang et al. [23] introduced a multi-dimensional storage structure with encrypted keywords (MSS-EK) tailored for sixth-generation mobile communication networks. Based on this framework, they proposed a secure and searchable edge precaching scheme. This scheme can intelligently precache requested data at edge nodes by utilizing the user’s location and directional information.

With the evolution of edge intelligence, industrial Internet, smart city, and related technologies, the scale of mobile edge computing networks is expanding rapidly, leading to increasingly dense deployments of base stations and edge servers. Consequently, the distribution of data from cloud servers to edge servers is consuming increasingly large amounts of energy. The substantial energy consumption costs incurred by data transmission pose a significant burden on communication operators [24]. Therefore, it becomes crucial to design a lowest energy consumption data distribution strategy that does not compromise user experience.

In this paper, considering the premise of not hurting the experience of end-users, we define the problem of minimizing energy consumption by distributing a data packet that needs to be cached in the mobile edge computing network from the cloud servers to the edge servers as the edge caching data distribution with minimum energy consumption (ECDDMEC) problem. Edge cache technology is predominantly employed in latency-sensitive scenarios where stringent network delay requirements are imposed. As a consequence, the time complexity of the devised edge data distribution algorithm must be kept low, ensuring prompt computation of results. The minimum k-Hop dominating set (MkHDS) problem stands as a quintessential NP-hard problem, predominantly applied in the realm of social networks in previous research [25,26]. In this paper, we make the first attempt to introduce the MkHDS problem to mobile edge computing networks. Subsequently, a greedy algorithm with a computational complexity of $O(n^{2})$ is devised to rapidly and approximately address the edge caching data distribution problem with the aim of minimizing energy consumption. The main contributions of this paper are as follows:We establish a system model for the edge caching system. Based on the system model, an ECDDMEC problem, which is involved with network latency, communication radius and energy consumption, is formulated.We prove the ECDDMEC problem to be NP-hard and use a greedy strategy to devise an approximation algorithm with a computational complexity of $O(n^{2})$ to solve it.Simulation results show that our proposed algorithm can reduce greatly the energy consumption associated with data distribution.

## 2. System Model

Figure 1 depicts the edge caching system model considered in this paper, comprising three components: the mobile edge computing network, cloud server, and mobile terminal users. In the mobile edge computing network, each network node is composed of a base station equipped with an edge server. The base station serves to provide communication services to users, while the edge server delivers computing and caching services to both users and application providers. *R* represents the communication radius of the base station. When the distance between two base stations is less than the communication radius, they can establish a communication link between each other. The communication link is represented by an undirected edge, named edge-to-edge (E2E) communication link. Furthermore, the cloud Server can communicate with every edge server via cloud-to-edge (C2E) communication links. Users situated in the network coverage area establish communication with the nearest base station. To ensure efficient and stable communication, the communication between different base stations employs independent communication channels.

In this paper, the network nodes within the mobile edge computing network that require caching of specific popular data are termed caching edge nodes, while the remaining network nodes are referred to as non-caching edge nodes. Caching edge nodes are denoted by the set $V=\left\{v_{1},v_{2},v_{3},\ldots ,v_{n}\right\}$. Since the terminal users are mobile, caching edge nodes are selected based on the terminal users’ geographical activity areas. Once an area is selected, all network nodes in the area are designated as caching edge nodes.

The mobile edge computing network depicted in Figure 1 is modeled as an undirected graph, as shown in Figure 2. Here, vertices in the graph represent network nodes comprised of base stations and edge servers, and edges in the graph denote communication links between base stations. In Figure 2, the presence of an edge connecting vertex 1 and vertex 2 indicates that base station 1 and base station 2 are within each other’s communication radius and can communicate directly.

## 3. Problem Formulation

In this sections, we propose and formulate the ECDDMEC problem as a constrained optimization problem and the notations for the problem are shown in Table 1.

The energy consumption associated with data distribution from cloud servers to cache edge nodes can be categorized into two components: firstly, the energy consumption during data transmission between cloud servers and edge servers, and secondly, the energy consumption resulting from data transmission among edge servers. Let $P_{C2E}$ denote the energy consumed by transmitting a unit bit of data from the cloud server to the edge server, and $P_{E2E}$ represent the energy consumed by transmitting a unit bit of data in a single hop between edge servers. Let *C* denote the size of the cache data packet. Therefore, the energy consumption of the data packet transmission from the cloud server to the caching edge node $v_{i}$, denoted as $E_{v_{i}}$, can be expressed as follows:(1)$$E_{v_{i}}=C\cdot \left(P_{C2E}+h_{i}\cdot P_{E2E}\right)$$
where $h_{i}$ represents the number of hops that a data packet is transmitted in the mobile edge computing network during the distribution process from the cloud server to the caching edge node $v_{i}$.

For instance, let us consider node 6 as a caching edge node, and the data transmission path is as follows: the data are initially transmitted from the cloud server to node 1, and subsequently relayed from node 1 to node 6 through node 2. In this scenario, the number of hops for data transmission in the mobile edge computing network amounts to 2, therefore, $h_{6}=2$.

Various application scenarios and data types impose distinct demands on data transmission delays. We define the latency of data transmission from the cloud server to the caching edge node $v_{i}$, denoted as $d(v_{i})$, subject to the constraint:(2)$$L\leq d(\nu_{i})\leq M$$
where *L* denotes the latency of data directly transmitted from the cloud server to the caching edge node $v_{i}$, and *M* represents the maximum tolerable latency when transmitting data from the cloud server to the caching edge node $v_{i}$ without compromising user experience.

Generally, in the same scenario, the latency for data transmission between edge servers in a mobile edge computing network remains consistent for each hop. Let *l* denote the latency for a single-hop transmission in the mobile edge computing network, thus:(3)$$d\left(\nu_{i}\right)=h_{i}\cdot l+L$$

Let *k* represent the maximum number of hops of data transmission from the cloud server to the caching edge node $v_{i}$ in a mobile edge computing network without compromising user experience, then:(4)$$L\leq d\left(\nu_{i}\right)=h_{i}\cdot l+L\leq k\cdot l+L\leq M$$

As both *L* and *l* are constants, for the sake of convenience of problem formulation and algorithm design, the latency of data transmission from the cloud server to the caching edge node $v_{i}$ is measured by $h_{i}$. Thus, the latency constraint can be expressed as:(5)$$0\leq h_{i}\leq k$$

In the mobile edge computing network, base stations communicate with each other through wireless communication links, while the cloud server communicates with base stations via wired communication links. Due to the limited communication radius of the base station, adjacent edge servers are in close geographic proximity. Conversely, in most cases, cloud servers and edge servers are considerably distant. Because of this, despite wired communication being energy-efficient, the energy consumption for transmitting the same data from the cloud server to the edge server remains substantially higher than that for transmitting a single hop in the mobile edge computing network. In this paper, we define a ratio β to represent the difference in energy consumption between them:(6)$$\beta =\frac{P_{C2E}}{P_{E2E}}$$

The mobile edge computing network, composed of caching edge nodes set $V=\left\{v_{1},v_{2},v_{3},v_{4},\cdots v_{n}\right\}$, can be simplified into an undirected graph G(V,E). The edges set *E* represents the communication links, and *n* represents the network size. Let node *c* denote the cloud server, and add node *c* into graph G(V,E). Then, connect node *c* with all nodes v∈V∖{c} to form a new graph $G_{new}$ and the strategy of data distribution from the cloud server to caching edge nodes *V* can be represented as a spanning tree $T_{d}$ generated by the graph *G* with *c* as the root.

During the data distribution process, caching edge nodes obtain data either from the cloud server or from adjacent caching edge nodes. In this paper, we designate the caching edge nodes that retrieve data from the cloud server as C2E caching nodes, represented by set $V_{C2E}$, and designate the caching edge nodes that obtain data from adjacent caching edge nodes as E2E caching nodes, represented by set $V_{E2E}$.

According to the properties of spanning trees in graph theory, for a given data distribution strategy $T_{d}$, the number of communication links $e_{c,v_{i}}$ from cloud servers to edge servers is equal to the number of C2E caching nodes $|V_{C2E}|$, and the number of communication links $e_{v_{i},v_{j}}$, between edge servers is equal to the number of E2E caching nodes $|V_{E2E}|$. Based on these properties, the energy consumed by distributing a data packet to all caching edge nodes using the data distribution strategy $T_{d}$ is given by:(7)$$\begin{array}{cc} E_{T_{d}} & =C\cdot \left(\left|V_{C2E}\right|\cdot P_{C2E}+\left|V_{E2E}\right|\cdot P_{E2E}\right)\\ \; & =C\cdot \left(\left|V_{C2E}\right|\cdot P_{C2E}+\left(n-\left|V_{C2E}\right|\right)\cdot P_{E2E}\right)\\ \; & =C\cdot \left(\left|V_{C2E}\right|\cdot \beta \cdot P_{E2E}+\left(n-\left|V_{C2E}\right|\right)\cdot P_{E2E}\right)\\ \; & =C\cdot P_{E2E}\cdot \left(\left|V_{C2E}\right|\cdot \beta +\left(n-\left|V_{C2E}\right|\right)\right)\\ \; & =C\cdot P_{E2E}\cdot \left(\left(\beta -1\right)\cdot \left|V_{C2E}\right|+n\right) \end{array}$$

In the process of data distribution, various strategies lead to varying energy consumption levels. To mitigate the energy consumption cost of communication operators, enhance network energy efficiency, and promote green and sustainable development in the communication industry, it is imperative to minimize data transmission energy consumption during the design of data distribution strategies. Therefore, in this paper, we consider minimizing $E_{T_{d}}$ which is the total energy consumption of distributing a data packet as the optimization objective expressed as follows:(8)$$\begin{array}{cc} \min E_{T_{d}} & =\min(C\cdot P_{E2E}\cdot (\left(\beta -1\right)\cdot |V_{C2E}\left|+n)\right)\\ \; & =C\cdot P_{E2E}\cdot n+C\cdot P_{E2E}\cdot \left(\beta -1\right)\cdot \min\left|V_{C2E}\right| \end{array}$$

From Equation (Equation 8), it is evident that the data distribution strategy with $|V_{C2E}|=1$ yields the lowest energy consumption. However, this strategy may potentially violate the delay constraint stated in Equation (Equation 5), thereby hurting the experience of end-users. Conversely, when $|V_{C2E}|=n$, the energy consumption is maximized. Let $E_{MAX}$ denote the energy consumption resulting from the data distribution strategy with $|V_{C2E}|=n$. $E_{MAX}$ can be expressed as follows:(9)$$E_{MAX}=C\cdot P_{E2E}\cdot \beta \cdot n$$

Based on the comprehensive analysis above, the ECDDMEC problem can be formulated as a constrained optimization problem as follows:

objective:(10)$$C\cdot P_{E2E}\cdot n+C\cdot P_{E2E}\cdot \left(\beta -1\right)\cdot \min\sum_{i=1}^{n}a_{i}$$

subject to:(11)$$\forall v_{i}\in V,\exists v_{j}\in V_{C2E},0\leq d\left(v_{i},v_{j}\right)\leq k$$

where:(12)$$a_{i}=\left\{\begin{array}{cc} 0 & \mathrm{if}\;v_{i}\notin V_{C2E}\\ 1 & \mathrm{if}\;v_{i}\in V_{C2E} \end{array}\right.$$

Equation (Equation 10) represents the optimization objective for the ECDDMEC problem. Constraint (11) denotes the latency constraint for data distribution, ensuring that any caching edge node $v_{i}\in V$ can reach a C2E caching node $v_{j}\in V_{C2E}$ within *k* hops, where $d(v_{i},v_{j})$ represents the hop count of the shortest path between node $v_{i}$ and node $v_{j}$. Constraint (12) is utilized to determine whether a caching edge node $v_{i}$ is a C2E caching node.

Upon solving the optimization problem, we obtain a minimum set of C2E caching nodes which is represented by $V_{C2E}^{*}$. Subsequently, we connect the cloud server node c to every node in the set $V_{C2E}^{*}$. Finally, we add edges *e* in set *E* to construct a tree, denoted as $T_{m}$. In this constructed tree $T_{m}$, all nodes in set $V_{E2E}$ are included, and the distance between the leaf nodes of the tree and root node *c* does not exceed k+1 hops. Hence, $T_{m}$ represents the data distribution strategy with the minimum energy consumption.

## 4. Problem Hardness

In this section, we will prove that the ECDDMEC problem is an NP-hard problem.

**Theorem** **1.**
*The ECDDMEC problem belongs to the class of NP problems.*


**Proof** **of** **Theorem** **1.**Given a data distribution strategy $T_{d}$, we design the following verification algorithm to verify if it is a solution to the ECDDMEC problem. The designed algorithm traverses all nodes $v_{i}$ in set *V* and verifies whether they adhere to constraints (11) and (12). Clearly, the computational complexity of this algorithm is O(n), which implies that it can run in polynomial time. Therefore, the ECDDMEC problem belongs to the class of NP problems.    □

**Theorem** **2.**
*The ECDDMEC problem is NP hard.*


**Proof** **of** **Theorem** **2.**To prove Theorem 2, we first introduce the MkHDS problem [25,26]. The MkHDS problem is a classic NP-hard problem, defined as follows [26]: Given an undirected graph G(V,E), the MkHDS problem aims to find a minimum vertex set $DS_{k}$ such that every vertex in the undirected graph G(V,E) either belongs to the set $DS_{k}$ or can be connected to at least one vertex in the set $DS_{k}$ through a path of no more than *k* edges. The mathematical model of the MkHDS problem is expressed as follows:Objective:
(13)$$\mathrm{Minimize}\sum_{v\in V}z_{v}$$Subject to:
(14)$$\sum_{v\in N(u,k)}z_{v}\geq 1,\;\forall u\in V$$where:
(15)$$z_{v}\in \left\{0,1\right\},\;\forall v\in V$$Here, $z_{v}$ is a binary variable indicating whether vertex *v* belongs to the *k*-hop dominating set, with $z_{v}=1$ if and only if $v\in DS_{k}$ is satisfied. N(u,k) denotes the set of neighbors of vertex *u* with no more than *k* hops in graph G(V,E).Suppose a popular data packet needs to be distributed to the edge servers in a certain region. Use an undirected graph G(V,E) to represent the mobile edge computing network in the region, where set $V=\left\{v_{1},v_{2},v_{3},v_{4},\cdots v_{n}\right\}$ represents all nodes in G(V,E). Let $T_{m}$ be the data distribution strategy obtained for solving the ECDDMEC problem. Clearly, the set of all neighboring nodes of the root *c* of $T_{m}$, the minimum C2E caching node set $V_{C2E}^{*}$ and the minimum k-hop dominating set of graph G(V,E) are equivalent.Based on the above analysis, when we obtain a solution $T_{m}$ for the ECDDMEC problem, we can find a minimum k-hop dominating set of graph G(V,E) using a polynomial-time algorithm. This algorithm is described as follows, with an algorithmic complexity of O(1): compute all neighboring nodes of node *c* of $T_{m}$. Therefore, the minimum k-hop dominating set problem can be reduced to an ECDDMEC problem in polynomial time. Since the minimum k-hop dominating set problem is NP-hard, the ECDDMEC problem is NP-hard.    □

## 5. Algorithm Design and Analysis

Given that the ECDDMEC problem is NP-hard, obtaining an exact solution in polynomial time is infeasible when the network size is sufficiently large. Therefore, in this section, we propose a greedy algorithm with a computational complexity of $O(n^{2})$, named ECDDMEC-A, to approximately and rapidly solve the problem in a large-scale mobile edge computing network.

### 5.1. Algorithm Description

ECDDMEC-A consists of three sub-algorithms: ECDDMEC-A-1, ECDDMEC-A-2, and ECDDMEC-A-3. ECDDMEC-A-1 is designed to construct a mobile edge computing network, denoted by an undirected graph G(V,E), based on a given set of caching edge nodes and the geographical location information of their base stations. ECDDMEC-A-2 aims to greedily and approximately find the minimum C2E caching nodes set $V_{C2E}^{*}$ for a given mobile edge computing network G(V,E). ECDDMEC-A-3 is devised to obtain the edge caching data distribution strategy aimed at minimizing energy consumption.

Algorithm 1 presents the pseudo-code of ECDDMEC-A-1 whose input is caching edge nodes set *V*, where each node is associated with its geographical coordinates, and the communication radius *R* of the base stations. Caching edge nodes are all mobile edge computing network nodes in the caching area which is determined based on the request information of data packages from end users.
**Algorithm 1** ECDDMEC-A-1: Construction of Mobile Edge Computing Network**Input:** caching edge nodes set $V=\left\{v_{1},v_{2},v_{3},v_{4},\cdots v_{n}\right\}$, geographical location information of the base station of each node in the set *V*, communication radius of the base station *R***Output:** G(V,E)  1: E←∅  2: **for** each node $v_{i}\in V$ **do**  3:    **for** each node $v_{j}\in V$ **do**  4:      calculate the Euclidean distance $D(v_{i},v_{j})$ between node $v_{i}$ and node $v_{j}$  5:      **if** $D(v_{i},v_{j})\leq R$ **then**  6:         Add $e_{v_{i},v_{j}}$ into *E*  7:      **end if**  8:    **end for**  9: **end for**10: Using node set *V* and edge set *E*, construct a graph G(V,E)11: **return** G(V,E)

ECDDMEC-A-1 begins by initializing an empty set *E* to store the edges of the graph (lines 1). Subsequently, it iterates over each pair of nodes in *V* to calculate the Euclidean distance $D(v_{i},v_{j})$ between them using their geographical coordinates (lines 4). If the distance between two nodes is less than or equal to the communication radius *R*, an undirected edge $e_{v_{i},v_{j}}$ which denotes a communication link between these two nodes will be added to set *E* (lines 5–7). After completing the iteration, the algorithm constructs an undirected graph G(V,E) using the node set *V* and the edge set *E* (lines 10). Finally, it returns G(V,E) as the output, representing the mobile edge computing network of the designated caching area (lines 11).

Algorithm 2 presents the pseudo-code of ECDDMEC-A-2 whose input is *k* and mobile edge computing network G(V,E). ECDDMEC-A-2 is a greedy algorithm starting with initializing an empty set $V_{C2E}^{*}$ to store C2E caching nodes (lines 1). Then, the algorithm iterates through a while loop (line 2–11) to approximately obtain the minimum C2E caching nodes set $V_{C2E}^{*}$. In each iteration of this while loop, the algorithm firstly traverses each node $v_{i}$ in *V* in a breadth-first search (BFS) order through a for loop (line 3–6) to compute the set $N(v_{i},k)$ and the number of elements of set $N(v_{i},k)$ which is denoted by $n\left(v_{i}\right)$. Subsequently, the algorithm randomly selects a node $u_{i}$ from set *U* which consists of nodes with the maximum neighbor nodes, within *k* hops, in the set *V* (line 7–8). Finally, the algorithm adds $u_{i}$ into set $V_{C2E}^{*}$ and removes $u_{i}$ and nodes in set $N(u_{i},k)$ from set *V* (line 9–10). When set *V* becomes empty, the while loop terminates and the approximate solution of minimum C2E caching nodes set $V_{C2E}^{*}$ is returned as the output.
**Algorithm 2** ECDDMEC-A-2: Solution of Minimum C2E Caching Nodes Set $V_{C2E}^{*}$**Input:** G(V,E), *k***Output:** minimum C2E caching node set $V_{C2E}^{*}$  1: $V_{C2E}^{*}\leftarrow \emptyset$  2: **while** 
V≠∅
 **do**  3:     **for** each node $v_{i}\in V$ in BFS order **do**  4:        Compute $N(v_{i},k)$  5:        $n\left(v_{i}\right)\leftarrow |N\left(v_{i},k\right)|$  6:     **end for**  7:     $U\leftarrow arg\max(n\left(v_{i}\right))$  8:     Randomly select a node $u_{i}$ from set *U*  9:     $V_{C2E}^{*}\leftarrow V_{C2E}^{*}\cup \left\{u_{i}\right\}$10:     $V\leftarrow V\setminus \{\left\{u_{i}\right\}\cup N\left(u_{i},k\right)\}$11: **end while**12: **return**
$V_{C2E}^{*}$

The pseudo-code of ECDDMEC-A-3 is presented by Algorithm 3. It takes the mobile edge computing network graph G(V,E) constructed by Algorithm 1, the minimum C2E caching nodes set $V_{C2E}^{*}$ obtained by Algorithm 2 and parameter *k* as input. The algorithm proceeds in three steps. In the first step, for each node $v_{i}$ in the set $V\setminus V_{C2E}^{*}$, the algorithm identifies the closest C2E caching node $s_{i}$ in the set $N(v_{i},k)$. This step establishes a mapping from the set $V\setminus V_{C2E}^{*}$ to $V_{C2E}^{*}$, which is stored using the function $dominating(v_{i})$. This mapping process is completed by the for loop from line 1 to line 4. In the second step, for each node $s_{i}$ in the set $V_{C2E}^{*}$, the algorithm constructs a tree $T_{s_{i}}$ with $s_{i}$ as the root, ensuring that each node $v_{i}\in V$ satisfying $s_{i}=dominating\left(v_{i}\right)$ belongs to $T_{s_{i}}$. This step is executed by the for loop from line 5 to line 12. In the third step, The algorithm initializes a tree $T_{m}$ with node *c* as the root and adds *c* into $T_{m}$. Then, by connecting node *c* to all nodes $s_{i}$ in set $V_{C2E}^{*}$, the algorithm links all trees $T_{s_{i}}$ together to construct a tree $T_{m}$ with node *c* as the root (lines 13–14). Finally, the algorithm returns the tree $T_{m}$ as the output denoting the edge caching data distribution strategy with minimum energy consumption (line 15). Data packets needing to be cached at edge servers are distributed from the cloud server following the topology of tree $T_{m}$.
**Algorithm 3** ECDDMEC-A-3: Obtaining Edge Caching Data Distribution Strategy with Minimum Energy Consumption**Input:** G(V,E), $V_{C2E}^{*}$, *k***Output:** $T_{m}$  1: **for** each node $v_{i}\in V\setminus V_{C2E}^{*}$ **do**  2:     Select a C2E caching node $s_{i}\in V_{C2E}^{*}$ in set $N(v_{i},k)$ with the minimum hop count from node $v_{i}$.  3:     $dominating\left(v_{i}\right)\leftarrow s_{i}$  4: **end for**  5: **for** each node $s_{i}\in V_{C2E}^{*}$ **do**  6:     Initialize a tree $T_{s_{i}}$ with node $s_{i}$ as the root: $T_{s_{i}}\leftarrow \emptyset$ and add $s_{i}$ into $T_{s_{i}}$  7:     **for** each node *v_i_* ∈ *N*(*s_i_*, *k*) in BFS order **do**  8:        **if** $s_{i}=dominating\left(v_{i}\right)$ **then**  9:            Add $v_{i}$ into $T_{s_{i}}$ with the minimum $d(v_{i},s_{i})$10:        **end if**11:     **end for**12: **end for**13: Initialize a tree $T_{m}$ with node *c* as the root: $T_{m}\leftarrow \emptyset$ and add *c* into $T_{m}$14: Add all tree $T_{s_{i}}$ into tree $T_{m}$ by connecting node *c* with node $s_{i}$15: **return** $T_{m}$

To intuitively display the execution process of the algorithm ECDDMEC-A, we take the mobile edge computing network shown in Figure 1 as an example, supposing k=2, and present a possible execution scenario of the algorithm ECDDMEC-A through an illustration shown in Figure 3.

Step 1: Select all nodes in the mobile edge computing network depicted in Figure 1 as caching edge nodes, and obtain geographical location information of each node which typically are the longitude and latitude of the base stations’ location. Then, by executing the sub-algorithm ECDDMEC-A-1, the mobile edge computing network is constructed as an undirected graph G(V,E), consisting of 12 vertices and 14 edges. The result is shown in Figure 3a.

Step 2: Execute sub-algorithm ECDDMEC-A-2 and obtain the minimum C2E caching nodes set $V_{C2E}^{*}$ of G(V,E). During the execution of ECDDMEC-A-2, The while loop (line 2 to line 11) iterates twice in total. In the first iteration, the algorithm traverses each node *v* in G(V,E), computes N(v,2) which is the set of neighbors of node *v* with no more than two hops, and selects a node with the maximum number of neighbors. As a result, node 1 is selected as the C2E caching node, as shown in Figure 3b. Finally, node 1 and its neighbors within two hops (nodes 2, 3, 4, 5, 6, 7, 8, 9) are removed. Now, only nodes 10, 11, and 12 remain in G(V,E) and next comes the second iteration. Clearly, all these three nodes satisfy the condition of having the maximum number of neighbors within 2 hops. Randomly choose a node (node 10) as a C2E caching node and the minimum C2E caching nodes set $V_{C2E}^{*}$ obtained by algorithm ECDDMEC-A-2 is 1,10, which is also the minimum 2-hop dominating set of G(V,E). The result is shown in Figure 3c.

Step 3: Obtaining two trees with node 1 and node 10 as roots, respectively. First of all, traverse each E2E caching node in graph G(V,E) and find their closest C2E caching node. This part is completed by executing lines 1 to 4 of the algorithm ECDDMEC-A-3. Clearly, node 1 is the closest C2E caching node to nodes 2, 3, 4, 5, 6, 7, and 8, and the closest C2E caching node to nodes 9, 11, and 12 is node 10. Specifically, node 6 has the same distance as nodes 1 and 10. Hence, either node 1 or node 10 can be chosen as the closest C2E caching node to node 6. Then, delete the edges of G(V,E) to obtain a tree whose root is node 1 consisting of nodes 1, 2, 3, 4, 5, 6, 7, and 8, and a tree whose root is node 10 consisting nodes 9, 10, 11, 12. This part is completed by executing lines 5 to 12 of the algorithm ECDDMEC-A-3. The execution result of step 3 is shown in Figure 3d.

Step 4: Add a node *c* representing the cloud server, and then add edges to connect node *c* with node 1 and node 10. This step is completed by executing lines 13 to 14 of the algorithm ECDDMEC-A-3. After executing lines 13 to 14 of ECDDMEC-A-3, a tree with node *c* as root is formed and the execution results are as illustrated in Figure 3e,f. The tree depicted in Figure 3f is the edge caching data distribution strategy with minimum energy consumption.

### 5.2. Algorithm Complexity Analysis

Algorithm 1 consists of a nested for loop with two levels, where each loop iterates over every node $v_{i}$ in the set *V* of the graph G(V,E). Therefore, the computational complexity is $O(n^{2})$.

Algorithm 2 mainly comprises a while loop from line 2 to line 11. During the execution of this algorithm, the while loop iterates $|V_{C2E}^{*}|$ times. Inside this while loop, there is a nested for loop. In the worst-case scenario, the for loop, from line 3 to line 6, iterates *n* times for each iteration of the while loop. Additionally, as the execution of line 4 of Algorithm 2 requires computation $\left|N(v_{i},k)\right|$ times for each execution, the computational complexity of each iteration of the while loop is $O(n\cdot |N\left(v_{i},k\right)|)$. Let $\bar{\left|N(v,k)\right|}$ denote the average value of |N(v,k)|. Based on the analysis above, the computational complexity of Algorithm 2 is $O(n\cdot \bar{\left|N(v,k)\right|}\cdot |V_{C2E}^{*}|)$. Since $O(\bar{\left|N(v,k)\right|}\cdot |V_{C2E}^{*}|)=O\left(n\right)$, the computational complexity of Algorithm 2 is $O(n^{2})$.

Algorithm 3 consists of two for loops. The first for loop, from line 1 to line 4, iterates over every node $v_{i}$ in the set $V\setminus V_{C2E}^{*}$, thus looping $|V\setminus V_{C2E}^{*}|$ times. In the worst-case scenario, each iteration of this for loop traverses all nodes in the set $N(v_{i},k)$, resulting in $\left|N(v_{i},k)\right|$ operations per iteration. The computational complexity of the first for loop is $O(\bar{\left|N(v,k)\right|}\cdot (n-|V_{C2E}^{*}\left|)\right)$, and since $O(\bar{\left|N(v,k)\right|}\cdot |V_{C2E}^{*}|)=O\left(n\right)$, the computational complexity of the first for loop is $O\left(\bar{\left|N(v,k)\right|}\cdot n-n\right)=O\left(\bar{\left|N(v,k)\right|}\cdot n\right)$.

The second for loop, from line 5 to line 12, iterates over every node $s_{i}$ in the set $V_{C2E}^{*}$, looping $|V_{C2E}^{*}|$ times. This for loop is a nested loop and its inner loop is another for loop that traverses each node $v_{i}$ in N(s,k), thus looping |N(s,k)| times. Its computational complexity is $O(\bar{\left|N(v,k)\right|})$. Similarly, since $O(\bar{\left|N(v,k)\right|}\cdot |V_{C2E}^{*}|)=O\left(n\right)$, the computational complexity of the second for loop is O(n).

The computational complexity of lines 13 to 14 of the algorithm is O(1). Therefore, the computational complexity of Algorithm 3 is $O\left(\bar{\left|N(v,k)\right|}\cdot n+n+1\right)=O\left(\bar{\left|N(v,k)\right|}\cdot n\right)$.

In summary, the computational complexity of the ECDDMEC-A algorithm is $O\left(2n^{2}+\bar{\left|N(v,k)\right|}\cdot n\right)=O\left(n^{2}\right)$; therefore, the algorithm can work out an approximate solution of the problem in polynomial time.

## 6. Experimental Evaluation

In this section, the following simulation experiments are designed to evaluate the performance of the algorithm in this paper.

### 6.1. Experimental Setup

The algorithms in this paper are implemented using the Python language. The simulation experiments are conducted on a computer system equipped with a 2.30 GHz Intel(R) Core(TM) i7-12700H CPU (Santa Clara, CA, USA), NVIDIA GeForce RTX 3070 Ti Laptop GPU (Santa Clara, CA, USA), and 16 GB RAM. The software used for the simulation experiments is Python 3.9, with the igraph, numpy, and matplotlib libraries employed for graph computation and data visualization.

Assume that the communication radius of the base stations is uniform and β is 20. For simplicity and objectivity, a network environment for the simulation experiment is obtained in the following approach: *n* base stations are uniformly distributed in a square area of 2km×2km. Each base station, along with an edge server, forms a node in the mobile edge computing network. In the experiments, the communication radius of the base stations is set to 100 m, 200 m, 300 m, 400 m, 500 m, and 600 m, respectively. The network size *n* varies from 100 to 2000 with an increment of 100. The maximum tolerable delay hop number *k*, which represents the maximum number of hops in the mobile edge computing network from the cloud server to the caching edge nodes, is a non-negative integer. Under the same set of parameters, we randomly produce simulation networks. To obtain an objective simulation result for each obtained simulation network, our proposed algorithm is executed 100 times to produce the results. Finally, we calculate their average value as the simulation results.

### 6.2. Performance Evaluation Metric

To evaluate the performance of the algorithms, this paper selects the energy consumption ratio γ as the performance metric, defined as follows:(16)$$\gamma =\frac{E_{T_{m}}}{E_{MAX}}$$
where $E_{T_{m}}$ represents the energy consumption of the edge caching data distribution strategy $T_{m}$ obtained by the algorithm proposed in this paper. Obviously, the smaller the energy consumption ratio γ, the better the performance of the proposed algorithm.

### 6.3. Experimental Results

Figure 4 illustrates the relationship between network size *n* and energy consumption ratio γ at different values of *k* when the communication radius of the base stations is, respectively, set to 100 m, 200 m, 300 m, 400 m, 500 m, and 600 m. It can be observed from Figure 4 that when the communication radius of the base stations and *k* are fixed, the energy consumption ratio of the algorithm decreases with the increase in network size *n*. Moreover, as the network size further increases, the energy consumption ratio eventually converges to approximately 0.05. Additionally, when the network size and the communication radius of the base stations are fixed, the energy consumption ratio decreases with the increase in *k*, and this decrease becomes more significant as the network size decreases.

Figure 5 presents the relationship between the communication radius of the base stations and the energy consumption ratio γ at different values of *k*, where the maximum data transmission delay *k* from the cloud server to the caching edge nodes is set to 1, 2, 3, 4, 5, and 6, respectively. Five representative network sizes are selected, including 100, 200, 300, 500, and 1000 network nodes. It can be observed from Figure 5 that when *k* and network size *n* are fixed, the energy consumption ratio γ decreases as the communication radius of the base stations increases. Furthermore, as the communication radius increases, the energy consumption ratio converges to approximately 0.05, and the convergence is faster to reach larger network sizes.

To explain the experimental phenomena mentioned above, we simultaneously combine Equations (Equation 7), (Equation 9) and (Equation 16), and expand the definition equation of the energy consumption ratio as follows:(17)$$\begin{array}{cc} \gamma & =\frac{E_{T_{m}}}{E_{MAX}}\\ \; & =\frac{C\cdot P_{E2E}\cdot n+C\cdot P_{E2E}\cdot \left(\beta -1\right)\cdot \left|V_{C2E}^{*}\right|}{C\cdot P_{E2E}\cdot \beta \cdot n}\\ \; & =\frac{n+\left(\beta -1\right)\cdot |V_{C2E}^{*}|}{\beta \cdot n}\\ \; & =\frac{1}{\beta }+\frac{\beta -1}{\beta }\cdot \frac{|V_{C2E}^{*}|}{n} \end{array}$$

From the above equation, we can observe that the energy consumption ratio γ comprises two components: the first component is $\frac{1}{\beta }$, and the second component is $\frac{\beta -1}{\beta }\cdot \frac{|V_{C2E}^{*}|}{n}$. When β remains constant, the value of γ is only related to the network size *n* and the number of C2E caching nodes $|V_{C2E}^{*}|$.

Figure 6 illustrates the relationship between the network size *n* and the number of C2E caching nodes under different values of *k*. From Figure 6, it can be observed that as the network size *n* increases, the number of C2E caching nodes tends to stabilize at a constant value, no longer varying with the increase in network size *n*. When the number of C2E caching nodes ceases to change, as the network size *n* continues to grow, the value of $\frac{\beta -1}{\beta }\cdot \frac{|V_{C2E}^{*}|}{n}$ approaches 0. At this point, the value of γ tends towards $\frac{1}{\beta }$, where β in this paper is set to 20. Therefore, γ converges to around 0.05 as the network size *n* increases, consistent with the experimental results shown in Figure 4.

When the network size and the communication radius of the base stations remain constant, with an increase in the maximum tolerable hop count *k* for data transmission from the cloud server to the caching edge nodes, the number of C2E caching nodes $|V_{C2E}^{*}|$ decreases. Consequently, the value of γ decreases. When $|V_{C2E}^{*}|$ decreases to 1, γ reaches its minimum value which is $\frac{1}{\beta }+\frac{\beta -1}{\beta }\cdot \frac{1}{n}\approx \frac{1}{\beta }$. Given that β is set to 20 in this study, γ converges to around 0.05, consistent with the experimental results depicted in Figure 4.

Similarly, when the maximum network latency *k* and the network scale *n* remain constant, an increase in the communication radius of the base stations results in a decrease in the number of C2E caching nodes $|V_{C2E}^{*}|$. Consequently, the energy consumption ratio γ decreases with the increase in the communication radius. When $|V_{C2E}^{*}|$ decreases to 1, γ reaches its minimum value of $\frac{1}{\beta }+\frac{\beta -1}{\beta }\cdot \frac{1}{n}\approx \frac{1}{\beta }$. Given that β is set to 20 in this study, γ converges to around 0.05, consistent with the experimental results depicted in Figure 5.

## 7. Conclusions and Future Work

To reduce the energy consumption cost of telecommunication operators and achieve the green and sustainable development of the communication industry, this study investigates the problem of minimizing energy consumption in edge caching data distribution in the context of mobile edge computing. This paper first formulates the problem and proves its NP-hardness. Subsequently, a greedy algorithm with a computational complexity of $O(n^{2})$ is designed to approximately solve this problem. Experimental results demonstrate that compared to the strategy of all edge servers directly fetching edge caching data from the cloud server, the proposed algorithm significantly reduces energy consumption during the data distribution process without compromising user experience. Moreover, in the same geographical area, the effectiveness of the proposed algorithm becomes more pronounced as the network size and the communication radius of the base station increase.

Utilizing artificial intelligence to predict the distribution and movement directions of users can facilitate the precise selection of caching areas and caching nodes. Precise selection of caching areas can not only reduce energy consumption during data distribution but also efficiently utilize the caching space of edge servers, which serves as a future research direction.

## Figures and Tables

**Figure 1 sensors-24-02898-f001:**
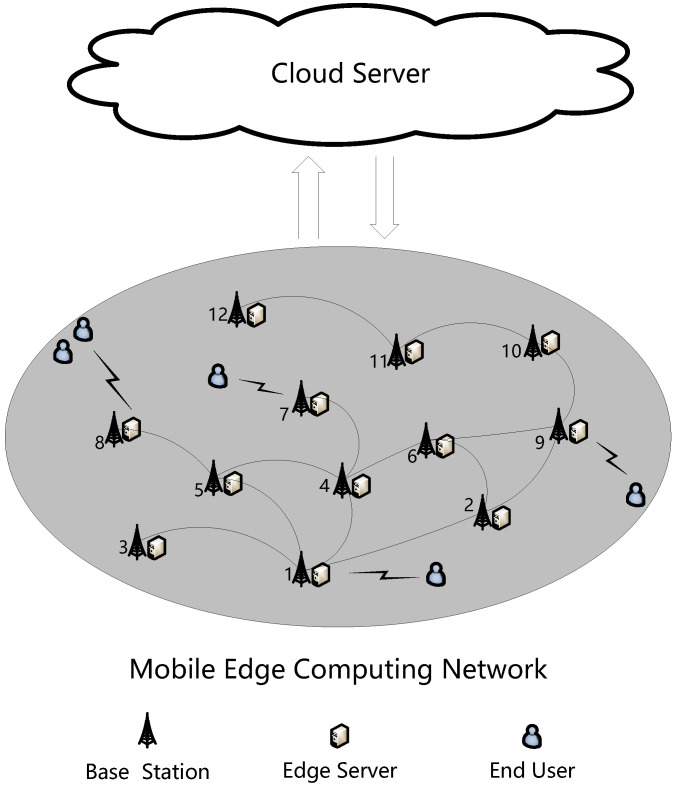
Edge caching system model Edge caching system model with 12 caching edge nodes labelled “1,2,⋯,12”.

**Figure 2 sensors-24-02898-f002:**
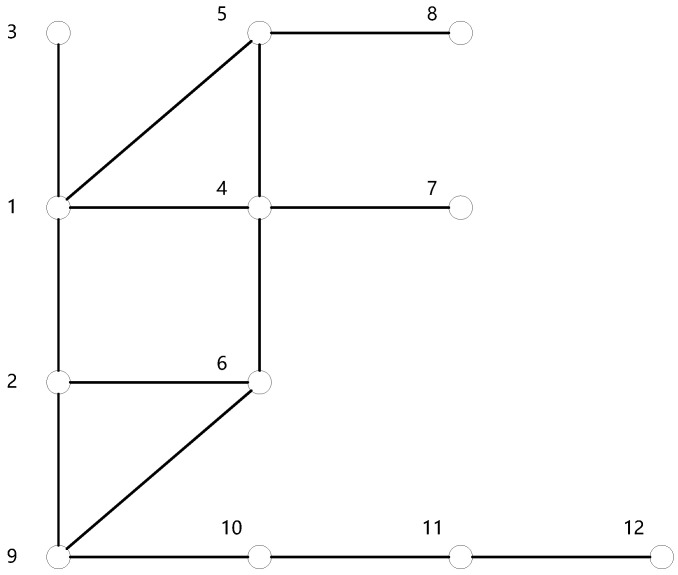
Undirected graph simplified from the mobile edge computing network in Figure 1.

**Figure 3 sensors-24-02898-f003:**
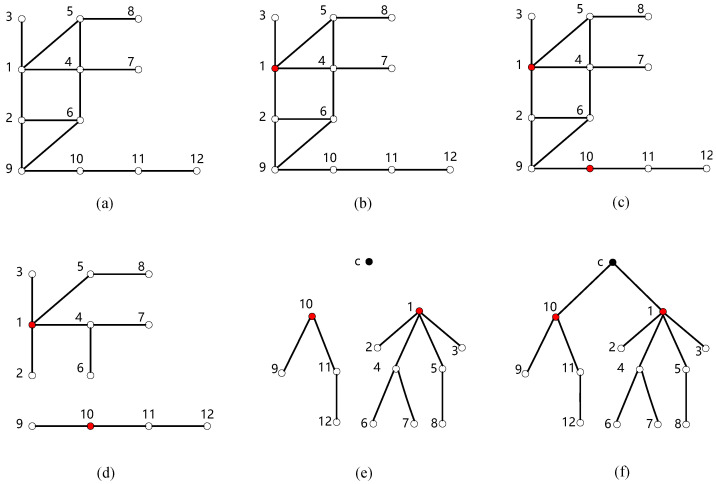
Execution process of the algorithm ECDDMEC-A consisting of the subgraphs (**a**–**f**).

**Figure 4 sensors-24-02898-f004:**
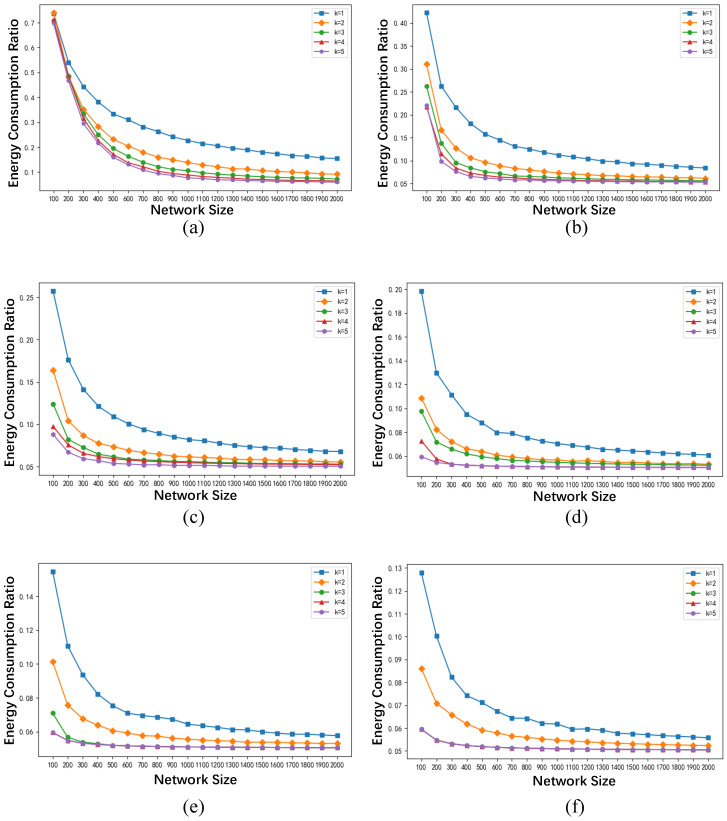
The relationship between network size and energy consumption ratio under different *k* conditions. (**a**) The communication radius of the base station is 100 m. (**b**) The communication radius of the base station is 200 m. (**c**) The communication radius of the base station is 300 m. (**d**) The communication radius of the base station is 400 m. (**e**) The communication radius of the base station is 500 m. (**f**) The communication radius of the base station is 600 m.

**Figure 5 sensors-24-02898-f005:**
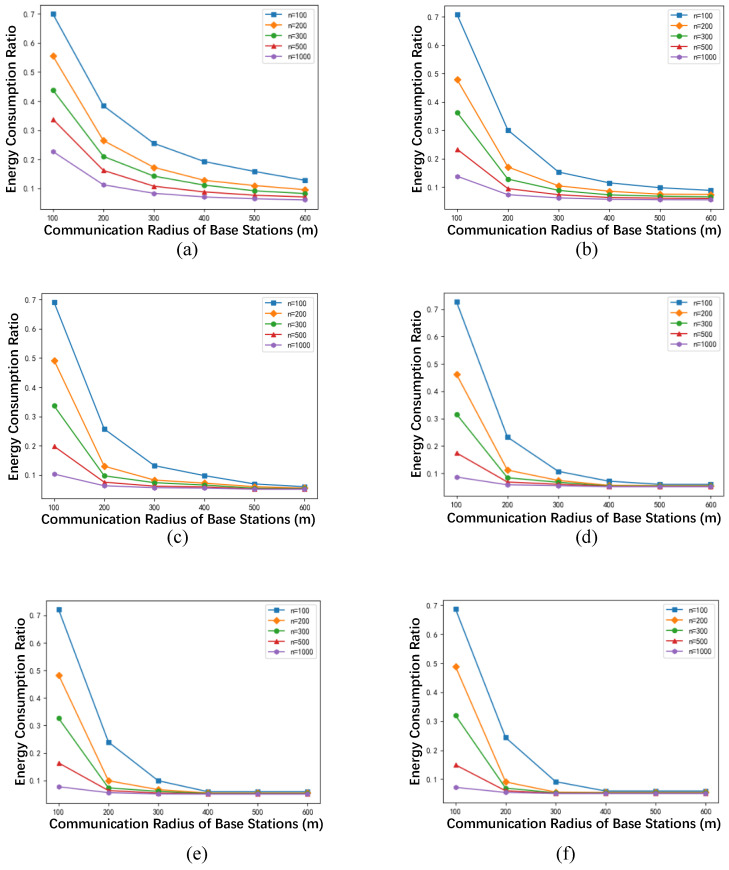
The relationship between base station communication radius and energy consumption ratio under different network sizes *n*. (**a**) k=1, (**b**) k=2, (**c**) k=3, (**d**) k=4, (**e**) k=5, (**f**) k=6.

**Figure 6 sensors-24-02898-f006:**
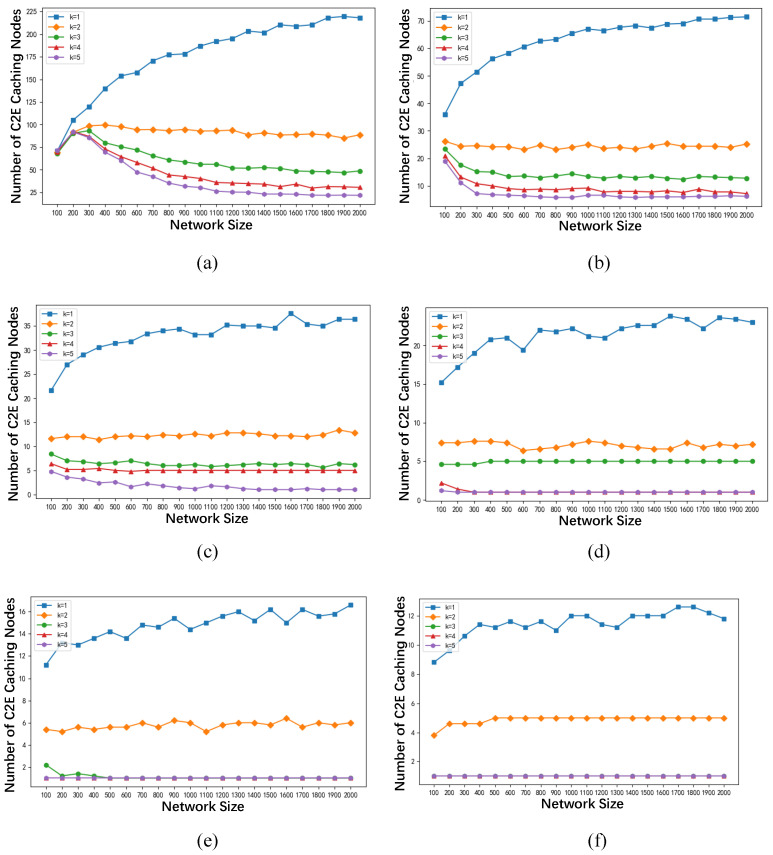
The relationship between network size and the number of C2E caching nodes under different *k* conditions. (**a**) The communication radius of the base station is 100 m. (**b**) The communication radius of the base station is 200 m. (**c**) The communication radius of the base station is 300 m. (**d**) The communication radius of the base station is 400 m. (**e**) The communication radius of the base station is 500 m. (**f**) The communication radius of the base station is 600 m.

**Table 1 sensors-24-02898-t001:** Summary of Notations.

Notation	Description
*R*	the communication radius of the base station
*c*	cloud server
$v_{i}$	caching edge node
*V*	set of caching edge nodes
*E*	set of communication links
$e_{v_{i},v_{j}}$	communication link between node $v_{i}$ and node $v_{j}$
*n*	network size
$E_{v_{i}}$	the energy consumption of the data packet transmission from the cloud server to the caching edge node $v_{i}$
*C*	the size of the cache data packet
$P_{C2E}$	the energy consumed by transmitting a unit bit of data from the cloud server to the edge server
$P_{E2E}$	the energy consumed by transmitting a unit bit of data in a single hop between edge servers
$h_{i}$	the number of hops that a data packet is transmitted in the mobile edge computing network during the distribution process from the cloud server to the caching edge node $v_{i}$
$d(v_{i})$	the latency of data transmission from the cloud server to the caching edge node $v_{i}$
*L*	the latency of data directly transmitted from the cloud server to the caching edge node $v_{i}$
*M*	the maximum tolerable latency when transmitting data from the cloud server to the caching edge node $v_{i}$ without compromising user experience
*l*	the latency for a single-hop transmission in the mobile edge computing network
*k*	the maximum number of hops of data transmission from the cloud server to the caching edge node $v_{i}$ in mobile edge computing network without compromising user experience
β	the ratio of energy consumption for transmitting the same data from the cloud server to the edge server remains and energy consumption for transmitting a single hop in the mobile edge computing network
$T_{d}$	strategy of data distribution from the cloud server to caching edge nodes
$T_{m}$	data distribution strategy with the minimum energy consumption
$V_{C2E}$	set of C2E caching nodes
$V_{E2E}$	set of E2E caching nodes
$V_{C2E}^{*}$	minimum set of C2E caching nodes
$E_{T_{d}}$	the total energy consumption by distributing a data packet to all caching edge nodes using data distribution strategy $T_{d}$
$E_{T_{m}}$	the energy consumption of the edge caching data distribution strategy $T_{m}$
$E_{MAX}$	the energy consumption resulting from the data distribution strategy with $|V_{C2E}|=n$
$a_{i}$	binary variable indicating whether $v_{i}$ is a C2E caching node
$d(v_{i},v_{j})$	the hop count of the shortest path between node $v_{i}$ and node $v_{j}$
$D(v_{i},v_{j})$	Euclidean distance between node $v_{i}$ and node $v_{j}$
N(u,k)	the set of neighbors of vertex *u* with no more than *k* hops

## Data Availability

Data is contained within the article.

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
