# Peer review of "Edge Caching Data Distribution Strategy with Minimum Energy Consumption"

_sensors, 2024, doi:10.3390/s24092898_

Round 1

Reviewer 1 Report

Comments and Suggestions for Authors

This research paper investigates the problem of minimizing energy consumption in edge caching data distribution in the context of mobile edge computing. Some of the comments that could improve the quality of this article are listed below.

- The contributions should be succinctly summarized to avoid fragmentation.

- The fonts in the figures are too small and less readable.

- Simulations currently involve the performance of the proposed algorithm under different parameter settings. It is recommended to add comparison with the latest algorithms.

- Too many short paragraphs in the introduction affect readability. The references are not particularly well cited and some of the references are out of date. The authors may think about including most recent work regarding "service intent-aware task scheduling" and “two-timescale hierarchical service deployment and task scheduling”into discussions.

- Proofreading is required to rectify any spelling errors in the article.

Comments on the Quality of English Language

This research paper investigates the problem of minimizing energy consumption in edge caching data distribution in the context of mobile edge computing. Some of the comments that could improve the quality of this article are listed below.

- The contributions should be succinctly summarized to avoid fragmentation.

- The fonts in the figures are too small and less readable.

- Simulations currently involve the performance of the proposed algorithm under different parameter settings. It is recommended to add comparison with the latest algorithms.

- Too many short paragraphs in the introduction affect readability. The references are not particularly well cited and some of the references are out of date. The authors may think about including most recent work regarding "service intent-aware task scheduling" and “two-timescale hierarchical service deployment and task scheduling”into discussions.

- Proofreading is required to rectify any spelling errors in the article.

Reviewer 2 Report

Comments and Suggestions for Authors

In this paper, the authors develop a system model for an edge caching system and address several critical factors such as network latency, base station communication radius, and data transmission energy consumption. The authors formulate the Edge Caching Delayed Data Mobile Edge Computing (ECDDMEC) problem, demonstrating that it is NP-hard. To tackle the ECDDMEC problem, the authors propose a greedy algorithm with a computational complexity of O(n^2), and provide an analysis of this algorithm's computational complexity. Additionally, the authors conduct a simulation experiment to evaluate the performance of their proposed algorithm. The results from this experiment confirm that their algorithm effectively enhances system performance.
However, several terms and aspects of this paper remain unclear or not well-explained so that the authors should take further modification or additional explanations:

1. The authors use a constant cache size value of C for formulating the edge caching problem. It is not clear why this value is constant. The reviewer suggests that the cache size might depend on the distributed data size, and typically, the amount of data content for caching in a mobile edge network is large. The rationale for using a constant value should be clarified.
2. The authors should include a summary of variable parameters in the form of a table. This would provide a clear and organized presentation of the parameters involved in the study, enhancing the paper's readability and comprehensiveness.
3. Furthermore, the authors should add an execution image for the proposed algorithm, illustrating some key parameters. This would help in visualizing the algorithm's workflow and understanding the interaction between different parameters.
4. The authors should describe the measurement scheme for all variable parameters in practical scenarios. Measuring parameters such as P_{C2E} and P_{E2E} is likely challenging due to their variability in the presence of background traffic. The paper should discuss how these measurements are approached under such conditions.
5. The evaluation environment described in the paper is unclear and does not adequately verify some use cases. The authors should provide details on the network topology used in their experiments. Additionally, it would be beneficial to verify the proposed algorithm in various scenarios, including changes in the network topology. This would demonstrate the robustness of the algorithm and its applicability to real-world conditions.

Comments on the Quality of English Language

No problem. But square character is in 279 rows in page 8. The author should remove it.

Round 2

Reviewer 1 Report

Comments and Suggestions for Authors

The authors have improved their paper, they have carefully answered to the reviewers' comments. It is a good piece of work and, in my opinion, it can be published.

Comments on the Quality of English Language

The authors have improved their paper, they have carefully answered to the reviewers' comments. It is a good piece of work and, in my opinion, it can be published.

Reviewer 2 Report

Comments and Suggestions for Authors

All modifications are correctly done.